# Mucosa–Environment Interactions in the Pathogenesis of Rheumatoid Arthritis

**DOI:** 10.3390/cells8070700

**Published:** 2019-07-10

**Authors:** Bruno Lucchino, Francesca Romani Spinelli, Cristina Iannuccelli, Maria Paola Guzzo, Fabrizio Conti, Manuela Di Franco

**Affiliations:** Rheumatology Unit, Department of Internal Medicine and Medical Specialities, Sapienza University of Rome, 00161 Rome, Italy

**Keywords:** rheumatoid arthritis, mucosal immunity, ACPAs, microbiota, lung, periodontitis

## Abstract

Mucosal surfaces play a central role in the pathogenesis of rheumatoid arthritis (RA). Several risk factors, such as cigarette smoking, environmental pollution, and periodontitis interact with the host at the mucosal level, triggering immune system activation. Moreover, the alteration of microbiota homeostasis is gaining increased attention for its involvement in the disease pathogenesis, modulating the immune cell response at a local and subsequently at a systemic level. Currently, the onset of the clinical manifest arthritis is thought to be the last step of a series of pathogenic events lasting years. The positivity for anti-citrullinated protein antibodies (ACPAs) and rheumatoid factor (RF), in absence of symptoms, characterizes a preclinical phase of RA—namely systemic autoimmune phase- which is at high risk for disease progression. Several immune abnormalities, such as local ACPA production, increased T cell polarization towards a pro-inflammatory phenotype, and innate immune cell activation can be documented in at-risk subjects. Many of these abnormalities are direct consequences of the interaction between the environment and the host, which takes place at the mucosal level. The purpose of this review is to describe the humoral and cellular immune abnormalities detected in subjects at risk of RA, highlighting their origin from the mucosa–environment interaction.

## 1. Introduction

Rheumatoid arthritis (RA) is a chronic, systemic autoimmune disease affecting mainly the joints. RA is characterized by synovial membrane inflammation, progressive cartilage damage, and disability [1]. Pathogenesis of RA is complex and heterologous. Several genetic and environmental factors concur to increase the risk of develop the disease. A series of pathological events occur before the clinical onset of the disease, in a preclinical phase that can last many years. The pre-RA is characterized by the emergence of autoantibodies in the absence of objective signs of joint inflammation [2]. Several studies, performed on sample collected before the onset of RA, showed that anti-citrullinated peptides antibodies (ACPAs) and rheumatoid factor (RF) are present up to 13 years before the clinical onset of the disease [3]. The presence of RA-related autoantibodies predicts the clinical development of the disease after a variable time lag, during which there is a progressive accumulation of multiple specificities of the autoantibodies, results of an epitope spreading phenomenon [4]. ACPAs and RF are considered hallmarks of seropositive RA and are biomarkers of a more erosive disease, greater radiographic damage, and worst prognosis [5]. A number of other antibodies have been identified in the serum of patients with RA, targeting post-translationally modified antigens as carbamylated or acetylated proteins [6,7]. Similarly to ACPAs, antibody anti-carbamylated proteins (Anti-CarP) have been detected years before the diagnosis and predict the evolution to RA [8,9]. While synovial membrane is certainly a source of autoantibodies in established RA [10], most of the subjects in the preclinical phase of RA show no evidence of joint involvement at histologic examination, even though their sera already contain autoantibodies [11]. The evidence of several immune abnormalities in subjects at risk for RA without any joint involvement suggests that the pathogenic events leading to the disease may occur outside the joints [12]. Many risk factors for RA are encountered at the mucosal level, so mucosal immunity alterations are supposed to play a central role in the pathogenic events preceding RA. Mucosae are the interface through which the host interacts with the environment. Virtually all the mucosal sites are equipped with barrier mechanisms, both physical and immunological, to protect the host from potentially harmful pathogens or toxins present in the environment. Apart from the epithelial surface—which represents a mechanical barrier to the environment—at the mucosal level, immunoglobulins, enzymes, complement, neutrophils, and macrophages act as a first line defense against environmental hazards [13]. Moreover, most mucosal surfaces possess a resident lymphatic tissue, the mucosal-associated lymphatic tissue (MALT), which can directly interact with antigens and develop immune responses. At the mucosal level, immunological events such as the development of antigen-specific T cells, B cell expansion, class switching, and local immunoglobulin production contribute to the development of the in situ immune response, which, through the migration of the antigen-carrying dendritic cells to the local lymph nodes, can induce a systemic immune response [14]. Besides the defensive function, the mucosal immune system also represents a site of interaction with commensal and symbiotic microorganisms, which together with the pathogenic microorganism represent the microbiota [15]. The variety of the microbiota composition shapes the immune response at the mucosal level, influencing both the innate and the adaptive immunity [16]. A perturbation of the normal microbiota homeostasis, through a reduction of protective bacterial species e/o an increase of potential pathogenic bacteria—i.e., the dysbiosis—has been implicated in several inflammatory and autoimmune conditions, including RA [17,18,19]. The purpose of this review is to explore the link between the immune abnormalities present in subjects at risk for RA and the possible influences of environmental factors at the mucosal level.

### A Multistep Model of RA Development and Genetic Predisposition

The current model of RA pathogenesis involves a multistep process, lasting decades before the clinical onset of the disease. The interplay between genetic and environmental factors predates the systemic immune alteration and contributes to the break of immunologic tolerance. Genetic predisposition is linked to more than 100 HLA and non HLA susceptibility loci, and overall 2/3 of the risk for the disease development is genetically determined [20]. The strongest genetic risk factor for RA lies within the human leucocyte antigen (HLA) class II region and encodes the HLA-DRB1 molecule. In particular, alleles encoding for the shared epitope (SE) are a strong risk factor for seropositive RA development. SE represents a motif of five amino acid residues in the binding groove of the DR chain, which can affect the antigen presentation to T cells [21,22]. Several non HLA loci have been associated with RA susceptibility, including PTPN22, CTLA4, and loci for cytokines, cytokines receptors, and signal transduction mediators [20]. In genetically susceptible individuals, the effect of several environmental factors may influence the development of systemic autoimmunity, such as the appearance of autoantibodies and the increase of circulating levels of cytokines and chemokines, in the absence of clinical manifestation [2]. Over time, the predisposed subjects may develop inflammatory arthralgia without a clinical arthritis. Patients with inflammatory arthralgia, positive for RF and ACPAs, have a high risk—nearly 30% in one year—of imminent arthritis development [23]. Recently, the European League Against Rheumatism (EULAR) released a definition of arthralgia suspicious for progression to RA, providing a set of clinical parameters to identify these patients for research purposes [24]. Although some patients with inflammatory arthralgia may show a subclinical synovitis, most of the patients do not present any sign of subclinical joint inflammation, suggesting that immune abnormalities might start outside the joint [25,26,27]. Abnormalities in the humoral and cellular components of innate and adaptive immunity are detectable in the systemic autoimmune phase of RA (synthetized in Figure 1).

## 2. Humoral Immunity Abnormalities in At-Risk Subjects

### 2.1. The Lung as a Site of ACPA Production

As mentioned above, the autoantibodies associated with RA can be detected years before the onset of the apparent disease. The lung is the most investigated site of autoimmunity generation, although the exact pathogenic interaction between gene and environment is still unclear. The strong association between cigarette smoke and ACPAs in subjects positive for SE-HLA-DRB1 alleles supports the mucosal origin of anti-citrulline response. SE-containing class II MHC molecules have a high affinity for citrullinated peptides, with consequent T cell presentation and autoimmune response to citrullinated self-antigens [28]. However, other mechanisms could explain the association between SE and RA. Recently, Fu et al. reported that SE can act as a calreticulin ligand and interact with the aryl hydrocarbon receptor (AhR), determining Th17 differentiation, osteoclast activation, and a worsening of experimental arthritis [29]. Indeed, AhR is the mediator of the response to air pollutants and is implicated in the pathogenesis of autoimmune disorders [30,31,32]. This intriguing mechanism could also explain the association between SE-containing HLA-DRB1 alleles and autoimmune disorders other than RA [33,34,35,36]. Besides the interaction with SE, cigarette smoke is a well-known inductor of citrullination. The conversion of arginine into citrulline, catalyzed by the different isoforms of the peptydil-arginine deiminase (PAD) enzymes, is a post-translational modification that is implicated in several physiologic processes and occurs nonspecifically in inflammatory conditions [37]. Indeed, there is an increased expression of citrullinated proteins and PAD2 in bronco-alveolar lavage (BAL) cells as well as in mucosal tissue from bronchial biopsy of healthy smokers compared to nonsmokers [38]. Lung mucosal exposure to air pollutants, including cigarette smoke, may be implicated in the generation of other self-antigens and consequently in the development of other autoantibodies associated with RA. Carbamylation is a proteins’ modification strictly associated with cigarette smoke exposure [39]. Carbamylated proteins have been demonstrated in the lung of patients affected by chronic obstructive pulmonary disease [39,40]. Besides tobacco smoke, several other respiratory inhalants have been associated with post-translational modifications. Exposure to silica as well as other inorganic dusts, such as in construction or electronic workers, have been associated with the risk of RA [41,42], acting synergistically with cigarette smoke [43]. Similarly, exposure to organic dust, such as textile [44], wood [45], and agriculture organic dust [46] has been associated with an increased risk for RA. Although the molecular mechanisms explaining the association of dusts with RA are far from being fully understood, several evidences link the exposition to inhalants to an increase in citrullination and to the induction of an inflammatory state in the lung. Cadmium (Cd) is a common contaminant of several organic and inorganic dusts, detectable also in cigarette smoke. Hutchinson et al. demonstrated that Cd nanoparticles induce the citrullination of cytokeratin in human bronchial epithelial cells through the activation of calcium channels followed by calcium influx in cells, which leads to PAD activation [47]. Moreover, Cd exposure is associated with an increased expression of calreticulin, which, as stated before, may explain the genetic risk associated with SE in inflammatory response predisposition [48]. The role of Cd in inducing citrullination may further explain the association between smoking, dust exposure, and ACPA-positive RA [49]. The induction of protein citrullination is a characteristic shared by nanoparticles of various origin—such as carbon or silicon nanoparticles—through an increase of calcium levels and PAD activation, before the onset of any inflammatory change [50,51]. In mice exposed to organic dust, Poole et al. detected B cell activation, inducible bronchus associated lymphoid tissue (iBALT) formation, inflammatory changes, and autoantibody production, as ACPAs and anti-malondialdehyde-acetaldehyde antibodies [52]. Diesel exhaust particles, one of the most important components of environment particulate emitted from diesel engines, can induce citrullination on human epithelia bronchial cells [53]. This observation could partially explain the epidemiologic association between the risk of RA and the high-density traffic pollution exposure [54].

The effect of inhalation pollutant, especially cigarette smoke, in HLA-SE positive subjects represents an example of gene–environment interaction, which plays a relevant role in the induction of seropositive RA. In seronegative RA, by contrast, smoking exposure plays a negligible role [55]. This evidence may thus suggest distinct etiologic origins of seronegative RA. Currently, T cell immune dysregulation seems to play a major role in the pathogenesis of seronegative RA [56]. Abnormalities in other mucosal sites, such as periodontal tissue and gut mucosa may play a more important role in these circumstances (see Abnormal T Cell Polarization and T Cell Phenotypes). The role of lung abnormalities in seronegative patients is currently not clear. A recent report failed to identify differences in the prevalence of various high-resolution chest tomography (HRCT) abnormalities between seropositive and seronegative RA patients, suggesting a possible involvement of lung compartments regardless of serologic status [57].

Lung mucosal immunity can be influenced by environmental exposure to pollutants or by change in microbiota homeostasis. Early RA patients show a lung dysbiosis characterized by a reduced presence of several bacterial genera such as *Actynomyces*, *Burkhordelia*, and periodontopathic taxa, including *Prevotella* and *Porphyromonas*, and an excess of *Pseudonocardia*. The presence of the genus *Prevotella* seems to correlate with the serum levels of IgA-RF and with the number of ACPA specificities, according to the role of this microorganism as a trigger of arthritis in both mouse models and humans [58,59,60]. Moreover, Scher et al. showed a similar dysbiosis in RA patients and a control group of patients with sarcoidosis, suggesting that inflammatory process per se may influence lung microbiota composition [58]. Similarly, Demoruelle et al. reported a lower prevalence of *Prevotella* genus and increased presence of *Haemophilus* and *Streptococcus* in subjects at risk of RA compared to healthy controls [61]. 

Considering the influence of the various environmental factors on lung mucosa, it is not surprising that the lung represents a site of autoantibody production. In fact, induced sputum of RA patients is rich in ACPAs compared to the serum, and subjects at risk for RA without detectable ACPAs in the serum can show positivity for ACPAs in the induced sputum [62]. Similar results have been observed on BAL from newly diagnosed, untreated RA patients, showing a local enrichment of ACPAs compared to serum; this latter study further suggests that the local production of autoantibodies may be an early event during the pathogenesis of the disease [63]. These results parallel the evidence of structural abnormalities in the lung of ACPA-positive subjects without RA as well as RA patients, further suggesting a subclinical lung inflammation as a driver of the local production of autoantibodies [64]. Indeed, ACPA production has been demonstrated in several other non rheumatologic conditions characterized by lung mucosal inflammation, as bronchiectasis or cystic fibrosis [65]. Inflammatory changes in airways and alveoli are strictly associated with seropositive RA. Reynisdottir et al. reported an almost double prevalence of parenchymal abnormalities in HRCT in ACPA-positive RA compared to seronegative RA and healthy controls [63]. Moreover, Reynisdottir et al. demonstrated bronchial biopsy inflammation characterized by ectopic lymphoid structures composed by (iBALT) with germinal centers and plasmacells in early ACPA-positive RA compared to seronegative RA [66]. Thus, iBALT may represent the mucosal structure responsible for the production of autoantibodies. Rangel-Moreno et al. detected citrullinated proteins as well as an increased neutrophil extracellular trap (NET) local release (see NETosis in Mucosal Inflammation and Citrullinated Antigen Generation) in iBALT, suggesting a role of iBALT in ACPA production [67]. Similarly, subjects at risk of RA show increased evidence of airway abnormalities in HRCT scans compared to healthy controls, mostly bronchial wall thickening, bronchiectasis, centrilobular opacities, and air trapping, independently of a previous history of smoking [68]. As reported above, the sputum of at-risk subjects may show positivity for ACPAs even in the presence of serum negativity, suggesting a local production of autoantibodies predating joint involvement [62]. Further studies demonstrated a temporal sequence in the ACPA specificity development at the lung level. Sputum antibodies anti-citrullinated fibrinogen, anti-citrullinated apolipoprotein E, and anti-citrullinated fibronectin are more prevalent in at-risk subjects compared to those with established RA, suggesting that these antigens may be the earliest targets of antibodies generated in the lung. Anti-citrullinated enolase antibodies were the only subset showing a concordance between serum and sputum in at-risk subjects, suggesting the appearance of these antibodies in the transition from mucosal to systemic autoimmunity phase. At last, sputum antibodies anti-citrullinated histones were the most prevalent at the serum level in both at-risk subjects and RA patients compared and healthy controls, suggesting that the appearance of these antibodies at the serum level may indicate a transition to the clinical manifest disease. Moreover, the number of sputum antibody specificities increases from the preclinical phase to the clinical one, suggesting a possible lung epitope spreading phenomenon—the same demonstrated at the serum level—which is concomitant to the transition to the clinical manifest disease [69].

### 2.2. Periodontal and Gut Dysbiosis in the Humoral Autoimmunity

Although the inflammatory changes at the mucosal site as drivers of the local production of autoantibodies have been mainly demonstrated at the lung level, other sites, such as periodontal tissue and the gut, may represent sites of generation of autoimmunity and sources of autoantibodies. Epidemiological studies demonstrated a strict relationship between chronic periodontitis (CP) and RA. The two diseases share many risk factors such as genetic predisposition (including HLA-DRB1), smoking, nutritional state, and low socioeconomic status [70,71]. Moreover, there is clinical evidence of an association between the two conditions and the treatment of periodontitis ameliorates the RA course [72]. Gingival tissues of patients with CP contain citrullinated proteins and express PAD2 and PAD4 [65]. ACPAs can be detected in gingival crevicular fluid and at the serum level in CP patients [65,73]. In RA patients, serum levels of carbamylated proteins have been associated with CP severity, suggesting that carbamylation may also occur in inflamed gingiva [74]. The severity of CP is associated with the positivity in the gingival crevicular fluid of RA-associated autoantibodies, even in non RA patients [75]. The severity of CP is also associated, in pre-RA subjects, to the subsequent risk of arthritis development [76] and to the ACPA positivity in first-degree relatives of patients with RA [77]. Therefore, CP may represent a triggering site, in genetically predisposed subjects, to systemic autoimmunity development. *Porphyromonas gingivalis* is a causative periodontal pathogen of CP. *P. gingivalis* is the only human pathogen known to express PAD and thus to be able to citrullinate bacterial as well as host proteins, in particular fibrinogen and enolase. Bacterial PAD, namely PPAD, shows also the unique feature of being able to citrullinate carboxy-terminal arginines of various peptides, not shared by human PAD: this mechanism can potentially expand the number of citrullinated epitopes that may be generated [78,79]. Other key virulence factors of *P. gingivalis* are gingipains, cysteine-proteases that act synergistically to PPAD cleaving peptides for subsequent citrullination, fimbriae, and lipopolysaccharide (LPS), which together activate a broad range of immune receptors, inducing an inflammatory response [80,81]. Thus, *P. gingivalis* endows both factors (inducing inflammation and providing PPAD) that can cause protein citrullination and local ACPA production at the gum level. Moreover, *P. gingivalis* is able to induce neutrophils NETosis (see NETosis in Mucosal Inflammation and Citrullinated Antigen Generation), which has been recently demonstrated to be gingipains-dependent [82]. Mikuls et al. detected higher levels of antibodies anti-*P. gingivalis* in RA patients compared to controls [83]. Similarly, a high prevalence of these antibodies has been demonstrated in pre-RA, correlating with RA-associated autoantibodies and to a high risk of RA; these evidences suggest that CP sustained by *P. gingivalis* may play a role in the earliest phase of the disease [84]. According to the role of oral dysbiosis in the pathogenesis of the CP [85], several pathogens have been linked to the disruption of local immune tolerance. *Aggregatibacter actinomycetemcomitans* (*Aa*), is a Gram-negative bacteria agent of CP and aggressive localized periodontitis [86]. Konig et al. demonstrated that crevicular fluid from CP patients contains extensively citrullinated proteins, mirroring the joint citrullinome of RA [87]. Among all the other CP-associated bacteria, *Aa* was the only one able to reproduce the pattern of citrullination detected in the RA joint, through the production of leukotoxin A (Ltx-A) [87]. Ltx-A is a pore-forming toxin able to increase the influx of calcium inside cells, to trigger hypercitrullination and to induce a neutrophil’s death similar to NETosis. Patients with RA, especially HLA-SE positive patients, showed a high prevalence of anti-Ltx-A antibodies, which were also associated with serum positivity of ACPAs and RF [87]. This last finding was, however, not confirmed in a different cohort [88]. Moreover, little is known about the interactions between the various potential pathogens in promoting arthritis. Oral inoculation with *Aa* or *Fusobacterium nucleatum* in a mouse model of collagen-induced arthritis resulted in a faster arthritis development when compared to a mix of these two bacteria plus *P. gingivalis*. While the reason for these different responses is not clear, Engstrom suggested that the different cytokine milieu arising in response to the pathogenic microbiota can influence the progression to arthritis [89]. Citrullination has been found in the periodontal tissue of subjects with CP, even independently by the presence of *Aa* or *P. gingivalis* in the gums, suggesting the importance of other bacterial species as well as the host PAD activity [90]. A study on recent-onset RA showed a different oral microbiota when compared to healthy controls, with a peculiar enrichment of the taxa *Prevotella* and *Leptotrichia.* Unusual bacteria such as *Anaeroglobus* correlated to serum levels of ACPAs and RF [59]. More recently, Chen et al. reported a different profile of oral microbiota of RA patients compared to osteoarthritis patients and healthy subjects, with an enrichment of eight bacterial species, including *Prevotella, Streptococcus, Porphyromonas, Haemophilus, Rothia, Actinomyces, Granulicatella, Leptotrichia, Lautropia,* and *Fusobacterium* in RA patients. Conversely, healthy subjects were selectively enriched of the phylum Firmicutes, Bacteriodetes, and Proteobacteria, which constitute the fundamental components of a homeostatic microbiota and drive immune-tolerant responses (see Abnormal T Cell Polarization and T Cell Phenotypes). The authors suggested that the abnormal oral microbiota profile could be a useful diagnostic tool [91]. Interestingly, some bacterial species may be directly implicated in several pathogenic aspects of the early RA [91]. For example, *Prevotella* shows the ability to remove galactose terminal residues from glycosylated immunoglobulins, increasing the inflammatory potential [92] (see Secretory Immunoglobulins and Aberrant Glycosylation).

Less is known about the role of gut mucosa in the pathogenesis of humoral abnormalities of RA, although several studies have highlighted the relationship between gut dysbiosis and RA [93]. Recently, a proteomic analysis on gut mucosa showed the presence of several citrullinated proteins both in RA patients and in healthy subjects. Among citrullinated proteins, 21 have been previously identified at lung and joint levels, and three of them (actin, vimentin and fibrinogen) are known ACPA targets. Colonic tissue from RA patients showed a relative abundance of citrullinated vimentin, suggesting that colon mucosa may play a role in the breaking of tolerance in RA [94]. 

Several recent studies described microbiota changes in RA patients. The dysbiosis of early RA is characterized by an increase of several species including *Lactobacillus, Prevotella, Clostridium, Gordonibacter, Eggerthella, Collinsella*, Actinobacteria, and Lacnospiracheae and by a decrease of Proteobacteria, Firmicutes, and Faecalibacterium [19,95,96,97]. Although gut dysbiosis is thought to influence the pathogenesis of RA, mainly causing an imbalance in T cell polarization and innate immune cell activation [98], some hypothesis link directly gut microbiota with the abnormal humoral response. *Prevotella* sp. gained attention for its implication in dysbiosis in several mucosal sites (see above). Other than driving a TH17 response and IL-6 and IL-23 production (see Abnormal T Cell Polarization and T Cell Phenotypes), *Prevotella* could be implicated in the anti-citrullinated immune response through a molecular mimicry mechanism. Recently, two new antigens, N-acetylglucosamine-6-sulfatase (GNS) and filamin A (FLNA), have been identified as targets of T and B cell responses in about half of RA patients [99]. GNS and FLNA presented by HLA-DR have a strong homology sequence with *Prevotella* sulfatase and other bacterial epitopes, potentially linking the immune reaction against the altered microbiota with the onset of a pathogenic systemic response, at least in a subgroup of RA patients. Moreover, Pianta et al. reported GNS citrullination in vivo and a correlation between antibody anti-citrullinated GNS and ACPA level [99]. Recently, an enrichment of intestinal *Prevotella* sp. has been demonstrated in subjects at risk for RA, suggesting an early development of gut dysbiosis [100]. Segmented filamentous bacteria (SFB) are commensal bacteria widely studied for their immune-modulating properties. Flannigan et al. demonstrated that SFB colonization has a potent effect in inducing Th17 differentiation, through the stimulation of IL-23 production in dendritic cells, which in turn can activate type 3 innate lymphoid cells (ILC3) to secrete IL-22 [101] (see Innate Lymphoid Cells May Link Microbiota Changes with T Cell Activation). The activation of IL-23/IL-22/IL-17 axis has been recently recognized as a key moment in the acquisition of the inflammatory activity of autoantibodies: in preclinical RA, this axis modulates the glycosylation pattern [102] (see Secretory Immunoglobulins and Aberrant Glycosylation). In collagen-induced arthritis, the depletion of gut microbiota after the induction of the arthritis reduced disease severity, the serum levels of cytokines, and the pathogenicity of anti-collagen antibodies through a change in the glycosylation pattern, but only slightly reduced the antibody serum levels. On the contrary, an early depletion of the microbiota, before the induction of the arthritis, was associated with a significant reduction of anti-collagen antibody serum levels and to a reduced arthritis severity. Those effects were also associated with a reduced intestinal production of IL-22 and IL-17A [103]. Accordingly, the activation of the IL-23/IL-22/IL-17 axis mediated by gut dysbiosis may take part in two different moments in the disease pathogenesis: after the induction of a systemic anti-citrullinated response, increasing ACPA pathogenicity through changes in glycosylation patterns, and before ACPA production, increasing the tendency to produce autoantibodies. Indeed, the last effect may be explained by the existence of a gut–lung crosstalk, in which gut dysbiosis contributes to the development of inflammation and autoimmunity at the lung level. At the gut level, SFB increases the polarization toward Th17, which can migrate to the lung, inducing iBALT formation and autoantibody production in the pre-arthritic phase. Moreover, SFB selectively expand autoreactive Th17 cells, co-expressing SFB-specific T cell receptors (TCRs) in addition to their self-reactive TCRs, further increasing the autoimmune process. These results highlight how perturbations in gut microbiota may drive the development of an autoimmune humoral response in remote sites and may represent the second hit in the transition to the clinically manifest disease [104]. However, data on the role of dysbiosis as a trigger of the systemic autoimmune response are not conclusive. Further studies will help in identifying the pathobiont species that may play a role in the various steps.

### 2.3. Secretory Immunoglobulins and Aberrant Glycosylation

The mucosal origin of the autoimmune reaction driving RA is suggested also by the relative abundance of the IgA response in at-risk subjects. ACPAs IgA in serum and sputum has been specifically associated with the future development of RA in at-risk patients [62,105]. Moreover, IgA isotype RF precedes the IgG response [106]. IgA is the most common isotype of immunoglobulin produced at the mucosal level. They can exist in two subtypes, a monomeric circulating form and a dimeric secretory form, though the latter can also be detected at a low concentration in the peripheral blood. The two dimers of the secretory IgA are bound to a peptide, the secretory component, which is the result of the cleavage of the receptor responsible for the lumen translocation of the Ig and can be detected in the peripheral blood [107]. Roos et al. described a strong association between cigarette smoke and secretory IgA-ACPAs, confirmed through the detection of the secretory component, suggesting a possible airway mucosal origin of the anti-citrullinated response [108]. Accordingly, seropositive subjects at risk for RA show an enrichment of circulating IgA plasmablasts, which share clonality with IgG plasma cells and recognize several RA-related antigens [109]. More recently, the secretory form of several autoantibodies—such as ACPAs, RF, and anti-CarP—have been reported to be increased in RA patient serum, to have a high specificity for the disease, and surprisingly to consist mostly of IgM [110]. This observation may be linked to changes in the microbiota composition. In fact, besides IgA, IgM can also be secreted at the mucosal level, where they play a key role in regulating the commensal bacteria population and host–microbiota interactions. Recently, Magri et al. described a population of memory IgM-B cells clonally related to IgM secreting plasma cells that inhabit gut mucosa [111]. These B cell populations rapidly produce secretory IgM in response to microbiota changes. The secretory IgM target several species of mucus-coated commensal bacteria, contributing together with IgA, to select a single species of commensal bacteria and to increase the diversity and the homeostasis of the microbiota itself [111]. Accordingly, the development of a gut dysbiosis may drive the development of autoreactive B cells at the mucosal level, thus justifying the increase in the secretory IgM directed against modified self-antigens.

Another alteration in the humoral immunity detectable before the onset of the disease is the aberrant glycosylation of the IgG molecules. Glycosylation is a post-translational modification that involves two potential sites on the IgG, one on the Fc portion and the other involving the Fab hypervariable region [112]. The glycosylation influences the biological properties of the IgG molecule, modulating the inflammatory activity. In fact, a reduced terminal galactosylation increases the complement-fixing activity of the IgG, while an increased fucosylation of the Fc region changes the affinity for the various Fc receptors [113]. A reduced terminal galactosylation and an increased core fucosylation of the Fc fragment have been reported in preclinical RA, and this modification precedes the onset of clinical arthritis. These modifications are prevalent in IgG with ACPA specificity [114,115]. In subjects at risk for RA, a reduced IgG-Fc terminal sialylation of ACPAs and a reduced sialyltransferase activity in plasmablasts have been reported [102]. The absence of sialic acid residues on Fc increases IgG inflammatory activities through an increased affinity for the FcγR [116,117]. Moreover, immune-complexes containing desialylated ACPAs stimulate osteoclastogenesis both in vitro and in vivo, increasing the inflammatory bone loss in RA patients, and the sialylation of the IgG abrogated the osteoclastogenic activity [118]. The reduced sialylation of IgG depends from IL-23 activated Th17 cells. In fact, Th17 accumulate in germinal centers in the asymptomatic phase and induce a downregulation of sialyltransferase in plasmablasts and plasmacells through an IL-21- and IL-22-dependent mechanism [102]. This pro-inflammatory reprogramming of Ig production could possibly drive the onset of the arthritis [102]. These previous observations may connect the increases pro-inflammatory properties of ACPAs, mediated by changes in the glycosylation pattern, with the imbalance of the T cell subtypes present in at-risk subjects. These events can in part arise from mucosal immunity alteration.

## 3. Cellular and Innate Immune Abnormalities

### 3.1. Abnormal T Cell Polarization and T Cell Phenotypes

To date, there are limited studies about the T cell subsets of at-risk subjects. In ACPA-positive subjects at risk for RA progression, a reduced frequency of naïve and regulatory T cells and an increased population of atypical T cells, hyper-responsive to TCR stimuli—i.e., the inflammation-related T cell—have been demonstrated in the peripheral blood [119,120]. The subset imbalance was predictive of arthritis progression. The reduced regulatory activity may be an early event in the pathogenesis of the disease. Reduced circulating and lymph nodes CD4+IL-10+ T cells have been reported in subjects at risk for RA, with a subsequent increase in peripheral CD4+IL-17A+ T cells in close relationship with the onset of the arthritis [121]. These results are in line with the observation of persistently increased levels of IL-17 and CD4+IL-17A+ T cells in peripheral blood of early arthritis patients [122]. Overall, this suggests a failure of regulatory activity in the asymptomatic phase, followed by the expansion of a Th17 autoreactive population, which could mediate the inflammatory manifestations. On the contrary, a previous study demonstrated an increase of circulating Th17 cells in at-risk patients and a decreased frequency of these cells in the peripheral blood of RA patients. In the same study, the authors detected an increased amount of Th17 in synovial tissue, indicating a selective homing of the Th17 cells in the inflamed sites [123]. The apparent discrepancy between these results could be explained by the timing of the subjects’ evaluation, considering that each individual may have differences in the various steps of disease progression. A multistep process in the Treg/Th17 balance seems to characterize the preclinical phase of RA, but the precise event sequence still needs to be clarified. The IL-10/INF-γ ratio produced by peripheral blood mononuclear cells (PBMCs) in response to citrullinated antigens show a progressive reduction from healthy subjects to preclinical RA and to RA; this highlights that the failure in the Treg control of autoimmunity may be an important event in the transition from preclinical RA to clinical arthritis [124]. Recently, Cheng et al. described a peculiar molecular signature in the peripheral blood of a subgroup of at-risk subjects [125]. This phenotype has been linked to specific C-to-T single-nucleotide polymorphisms (SNPs) located at position 1858 (C1858T) of the human protein tyrosine phosphatase PTPN22 gene [125]. The role of PTPN22 as a risk locus for RA is well known, as it is the non HLA locus with the highest risk for RA, additive to the risk-related HLA alleles [55,126,127]. In fact, PBMCs of at-risk subjects show an increased level of citrullinated histones, an increased production of IL-2 and Th17 cytokines, and a reduced production of Th2 cytokines; moreover, T lymphocytes have a peculiar hyperproliferative, hyperinflammatory, and hypoglycolytic phenotype [127]. The attenuated phosphatase activity of PTPN22 strengthens the activation signals in lymphocytes and reduces the expression of glycolytic enzymes, explaining the increased production of IL-2 and the hypoglycolytic phenotype [127]. A reduced glycolysis in T cells is associated with a depletion of intracellular ROS through an increased shift of substrates toward the pentose phosphate pathway, which promotes the generation of NADPH and glutathione [128]. The lack of oxidant signaling drives T cells through a hyperproliferative and hyperinflammatory phenotype, and restoring the redox balance can suppress the pro-arthritogenic effector function in RA [129]. Moreover, PTPN22 is a critical regulator of PAD4, performing an inhibitory effect independent from its phosphatase activity. Attenuation of the non phosphatase activity of PTPN22—inducing hypercitrullination—has been related to the aberrant expression of Th17 and Th2 cytokines. In fact, the abnormal cytokines profile could be normalized by a pan-PAD inhibitor, although the precise mechanism explaining how hypercitrullination causes the aberrant expression of Th2 and Th17 cytokines is still unclear [129].

Alteration in mucosal microbiota plays a dominant role in shaping T cell polarization. An homeostatic, healthy, and heterogeneous microbiota, through the production of short chain fatty acids, induces the production of transforming growth factor-β in dendritic and epithelial cells, promoting the differentiation of T naïve cells in Treg [130]. On the contrary, several pathobionts that dominate RA dysbiosis can alter the Th17/Treg balance. *Prevotella copri* induces dendritic cell-mediated naïve T cell differentiation through Th1 and Th17 [97,131]. A relative abundance of *Prevotella* correlates with a reduction of probiont species (such as *Bacterioides fragilis*); *Bacterioides fragilis* exerts an anti-inflammatory activity by increasing the Tregs through the interaction between Toll-like receptor 2 (TLR2) and polysaccharide A [19,132]. As stated before, SFB induce a strong increase of Th17 cells. Recently, a human gut commensal bacteria, *Bifidobacterium adolescentis*, replicated the same effects of SFB in mice, thus possibly playing a role in RA-associated dysbiosis [133]. Actinobacteria, specifically *Collinsella*, are increased in the gut of RA patients and can trigger inflammation through an enhanced IL-17 production from epithelial intestinal cells [134]. The influence of dysbiosis at the gut level on T cell polarization are in part mediated from the activation of local innate immune cells (see Innate Lymphoid Cells May Link Microbiota Changes with T Cell Activation). The loss of the protective immunological function of the vermiform appendix may also contribute to the imbalance of the bacterial species, to the dysbiosis persistence, and to the pro-inflammatory state. The vermiform appendix represents a highly immunologic organ particularly rich of Treg, contributing to the peripheral tolerance. This tolerogenic function may be related to the particular local microbiota composition of the vermiform appendix. In fact, the most abundant bacteria phyla found in the mucosal biofilm of the vermiform appendix are *Firmicutes, Bacteroidetes, Actinobacteria*, and *Proteobacteria*, largely confirming the role of the appendix as a commensal bacteria reservoir. Consequently, acting as a probiont species niche, the appendix may contribute to microbiota homeostasis re-inoculating a large bowel of commensal bacterial species after intestinal perturbations or infections, following diarrheal clearance [135]. Epidemiological studies reported an association between the increased risk of RA and a previous appendicectomy that may be partially explained by the loss of the appendix contribution to the microbiota homeostasis [136,137]. However, other studies failed to confirm this association, and more investigations are needed to understand the role of the appendix in the microbiota homeostasis maintaining and in RA pathogenesis [138,139].

Periodontitis is also associated with a strong differentiation in terms of Th17. Oral microbiota influences the accumulation of Th17, the conversion of Treg in Th17, and local and systemic pro-inflammatory cytokine production [140]. *P. gingivalis* contributes to Th17 imbalance, promoting their differentiation through gingipains activity, increasing the local levels of Th17 and pro-inflammatory cytokine production [140].

### 3.2. Innate Lymphoid Cells May Link Microbiota Changes with T Cell Activation

Innate lymphoid cells (ILCs) are a population of lymphoid origin, which lack of somatically rearranged antigen receptors. Three main subtypes of ILCs are involved in shaping the immune response. ILC1 and ILC3 are mainly inflammatory subsets, producing INF-γ and IL-17, respectively [141]. An increased frequency of ILC1 is present at the lymph node level in RA patients as well as in at-risk subjects, while ILC3 is increased in patients with early RA [142]. Although the precise dynamics of these changes remain unclear, the increased prevalence of pro-inflammatory subsets of ILC in early phases of the disease suggests a role of these populations in the imbalance toward Th17 and Th1 differentiation, which may originate at the mucosal level. Indeed, ILC3 are a key cell type in the regulation of microbiota composition. These cells can directly respond to several microbial components through TLR2, -3, and -9. AhR is a key regulator of these cells, sensing the tryptophan catabolite indole-3-carbinol produced by various bacterial species—notably *Lactobacillus*—and inducing IL-22 production. Moreover, ILC3 also responds indirectly to several cytokines, such as IL-23 and IL-1β, produced by mononuclear phagocytes under the control of microbial signals. The main products of ILC3—i.e., granulocyte-macrophage colony stimulating factor (GM-CSF), IL-17, TNF, and IL-22—aside from controlling the epithelial barrier integrity and the production of epithelial antimicrobial peptides, enhance Th17 activity through the production of serum amyloid A [143,144]. The microbiota-induced activation of ILC3, together with the stimulation of other resident innate immune cells, such as dendritic cells and macrophages, directly influences the locally produced cytokines milieu and drives Th17 differentiation. Several pro-inflammatory cytokines, such as IL-6, IL-1β, and IL-23, stimulate the differentiation of naïve T lymphocytes through Th17 phenotypes, which in turn may increase the inflammatory response at the systemic level, worsening or triggering joint inflammation [145] (see Figure 1). Accordingly, Schirmer et al. demonstrated how systemic inflammatory cytokine production is influenced by microbiota composition. Using metagenomic sequencing to identify the taxonomic profile of stool samples, the authors demonstrated that microbiota composition directly correlates with cytokines produced by stimulated PBMC, therefore influencing the systemic cytokines response [146].

### 3.3. NETosis in Mucosal Inflammation and Citrullinated Antigen Generation

Neutrophils are involved in the mucosal pathogenesis of RA. These cells represent the first-line defense against pathogens, migrating to inflammatory sites under the effect of chemotactic stimuli and providing antimicrobial activity through several mechanisms such as phagocytosis, degranulation, and oxidative bursts [147,148]. A relevant defense mechanism involved in RA pathogenesis is the neutrophil death by NETosis. Neutrophil extracellular traps (NETs) are strands of nucleic acids bound to nuclear, cytoplasmic, and granular proteins extruded in the extracellular space induced by several microbial and non microbial danger signals [149]. The NETosis process requires the activity of PAD4, which, citrullinating the histones, promotes chromatin decondensation [150]. Thus, the extracellular release of NET represents a source of citrullinated autoantingens, such as histones, and other proteins, such as vimentin [151]. Citrullinated molecules released during NETosis are targets of ACPAs in RA; ACPAs in turn can stimulate NETosis [151]. Antibodies directed against citrullinated histones are present in up to 67% of patients with established RA and up to 48% in early RA [152]. These antibodies are also present in a subgroup of at-risk subjects, and their serum levels and their presence increase steadily over time before the onset of joint symptoms [153]. Citrullinated histones and their specific autoantibodies may appear early in the pathogenesis of the diseases. In fact, Chang et al. demonstrated that neutrophils from a group of RA patients show a heightened propensity for spontaneous NETosis and hypercitrullination. This phenotype is associated with C1858T SNPs of the PTPN22 gene [125]. As mentioned above, non phosphatase activity of PTPN22 is a critical regulator of PAD4. The increased activity of PAD4 leads to higher citrullination and spontaneous NETosis, which, in the presence of susceptible HLA alleles, may induce the development of ACPAs [125]. Considering that the increased propensity to NETosis related to PTPN22 C1858T SNP has been associated with several autoimmune diseases, the specific role of PAD activities in the regulation of immune cells in RA is yet to be defined [149].

Increased neutrophil NETosis has been identified in several mucosal sites potentially implicated in RA pathogenesis. In periodontitis, there is a local production of citrullinated proteins and ACPAs as well as increased NETosis [154]. Causative periodontitis agents strongly induce NETosis. Gingipains produced by *P. gingivalis* are potent activators of NET release, through the proteolytic activation of the PAR-2 receptor. NET induced by *P. gingivalis* lack bactericidal activity for the subsequent proteolytic inactivation of the NET proteins by gingipains. These mechanisms turn NET in favor of *P. gingivalis,* generating an environment rich in peptides and growth factors released by dying neutrophils that sustain the proliferation of pathogens [82]. Ltx-A produced by *Aa* is another potent inducer of leucotoxic hypercitrullination, a toxin-mediated cell death similar to NETosis. Thus, the generation of citrullinated proteins may explain the correlation between anti-Ltx-A antibodies and serum ACPAs detected in RA patients [87,155]. NETs are abundant in the gingival crevicular fluid of patients with CP [156]. Moreover, levels of circulating NET correlate with periodontitis severity in RA patients [74].

A number of evidences support the role of NETosis in the local inflammation of several lung diseases [157]. Moreover, cigarette smoke triggers NETosis directly or as an effect of nicotine, through the activation of acetylcholine receptors, and consequently induces the activation of dendritic cells and Th17 polarization [158,159]. Nicotine administration increases NETosis and worsens inflammation in arthritis mouse models [159]. Subjects at risk for RA show an increase of NET in the sputum, parallel with the sputum ACPA level, suggesting a role for NETosis in local ACPA production [160]. A recent study showed a strong correlation between sputum NET and ACPAs directed towards several citrullinated proteins, including proteins belonging to neutrophil cargo [69]. Besides generating citrullinated peptides, NET may also be implicated in the development of iBALT. In fact, neutrophil elastase contained in NET can convert IL-33 to its active form [161], triggering the release of IL-17F by bronchial epithelial cells [162]. IL-17 trapped in NET then further recruits inflammatory cells locally. IL-17 is a potent trigger for iBALT formation, acting together with the stroma-stimulating effect mediated by NETs [163]. Accordingly, under the influence of environmental factors, NETosis may be another trigger of local immune activation in lung mucosa.

## 4. Conclusions

The understanding of molecular events occurring before the onset of RA has greatly improved. The role of the mucosal abnormalities is relevant in the pathogenesis of the disease, with a determinant contribution of each of the most implied mucosal sites. A series of pathogenic events may outline a possible model of systemic autoimmune response development, in which oral and lung mucosa, under the stimuli of environmental factors, represent sites of ACPA production while gut dysbiosis increases the inflammatory state though increased Th17 polarization and IL-23/IL-17 axis activation (Figure 2). ACPAs play a central role in the progression of the disease. The recent discovery of the effect of ACPAs on osteoclastogenesis and on peri-articular IL-8 production suggests a mechanism explaining the transition from systemic autoimmunity to clinical symptoms, opening to the possibility of therapeutic interventions to prevent the development of the disease [164]. However, targeting B cells and consequently ACPA production through rituximab only delayed the diagnosis, instead of preventing it [165]. The failure of the PRAIRI trial, as well as other preventive strategies, suggests that, to achieve an effective and preventive strategy, we should focus on different pathogenic mechanisms [166]. Mucosal abnormalities are interesting targets for therapeutic strategies. Increasing the insights of the molecular events at this level in at-risk subjects may offer new possibilities. For example, increasing evidence of the role of IL-23/IL-17 may suggest a preventive strategy acting on this axis. A better characterization of mucosal immunity and its role in the pathogenesis of RA is an open issue and should be a priority in the research agenda.

## Figures and Tables

**Figure 1 cells-08-00700-f001:**
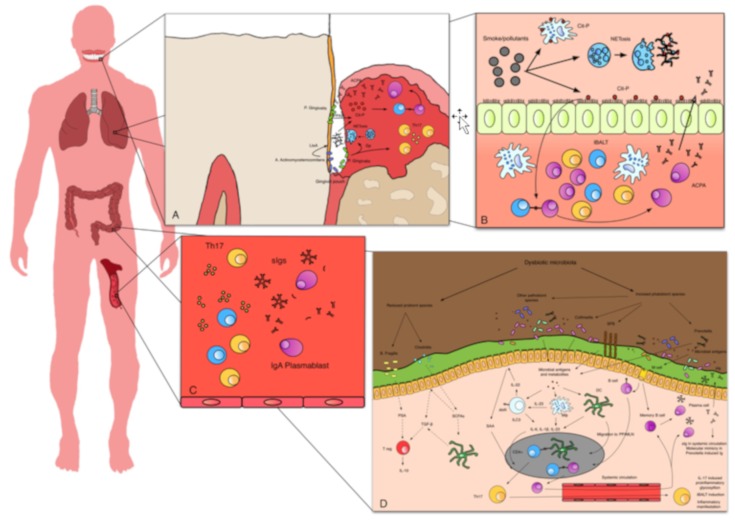
Immune abnormalities in at-risk subjects: (**A**) At the periodontal level, *P. gingivalis* generates citrullinated peptides through PPAD. Moreover through gingipains, *P. gingivalis* increases the polarization versus Th17 and induces NETosis. Aggregatibacter can also induce the formation of citrullinated peptides, through the production of Lxt-A and the induction of NETosis. Citrullinated peptides, recognized by specific B cells, induce ACPA production. (**B**) In the lung mucosa, smoke and air pollutants induce the formation of citrullinated antigens and NETosis. Mucosa reacts through the formation of iBALT and the local production of ACPAs, which can be detected in local secretions. (**C**) In systemic circulation, there is an increase of circulating Th17 and a reduction of Treg. T cells present an abnormal hypoglycolytic and hyperproliferative phenotype, and there is an increased production of pro-inflammatory cytokines, such as IL-17. Moreover, there is an enrichment of circulating IgA plasmablasts, as well as secretory IgA and IgM. (**D**) In gut mucosa, dysbiotic microbiota limits the normal induction of Treg. Pathobiont species stimulate the activation of DC, macrophages and ILC3, the polarization towards Th17, and the activation of the IL-23/IL-17 axis. Locally produced Th17 can migrate through systemic circulation in other sites, inducing inflammation, abnormal Ig glycosylation, and iBALT formation. Specific B cells directed against luminal antigens can be activated in Peyer’s patches or in local lymph nodes, migrating back in lamina propria where they can produce secretory Igs. Some of these B cells recognize antigens that, through molecular mimicry, cross react with self-antigens. Cit-p: citrullinated proteins; PPAD pathogen PAD; Gp: gingipains¸Ltx-A: leukotoxin A; iBALT: inducible bronchus associated lymphatic tissue; sIgs: secretory immunoglobulins; PSA: polysaccharide A; SCFAs: short chain fatty acids; SFB: segmented filamentous bacteria; AhR: aryl hydrocarbon receptor; SAA: serum amyloid A; ILC3: innate lymphoid cells 3; DC: dendritic cells; Mφ: macrophage; IL-: interleukin-; TGF: transforming growth factor; PP: Peyer’s patches; MLN: Mesenteric lymph nodes.

**Figure 2 cells-08-00700-f002:**
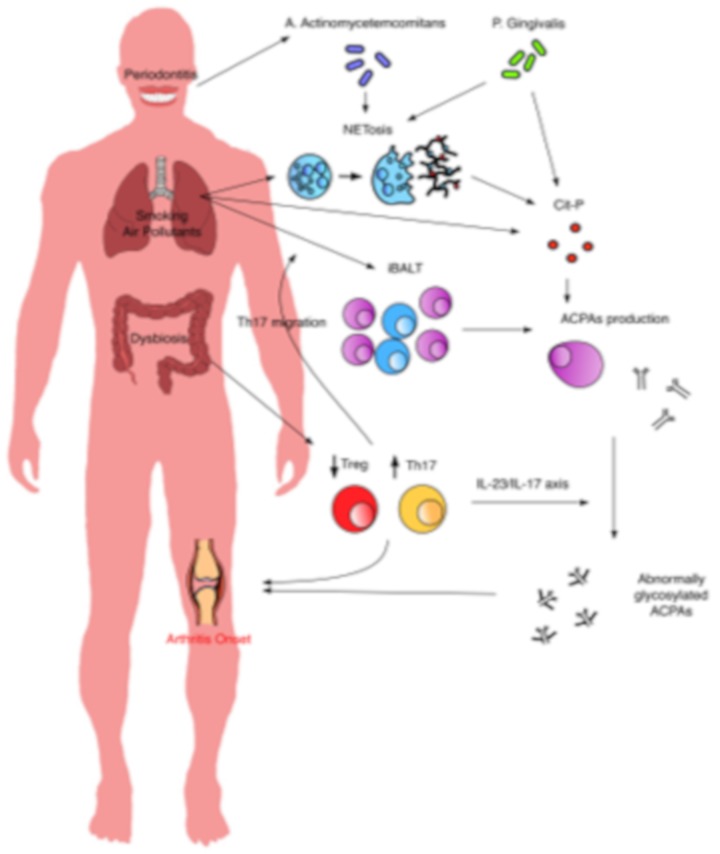
Interplay between mucosal events. At oral and lung levels, under the stimulus of periodontal pathogens or environmental factors, there is a production of citrullinated proteins, directly or through NETosis, and consequently ACPA production. At the gut level, dysbiosis induces the imbalance between Th17 and Treg. Th17 may migrate to the lung, contributing to iBALT formation, and at the systemic level, where, through the IL-23/IL17-axis, the abnormal glycosylation and the acquisition of ACPA pro-inflammatory activity are triggered. Inflammatory cells and ACPAs can at last induce the onset of arthritis. Cit-p: citrullinated proteins; iBALT: inducible bronchus associated lymphatic tissue; IL-: Interleukin-.

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
