# Peer review of "Mucosa–Environment Interactions in the Pathogenesis of Rheumatoid Arthritis"

_cells, 2019, doi:10.3390/cells8070700_

Round 1
Reviewer 1 Report
A nicely written narrative review.
While the authors have elected to focus on a mucosal enviroment type of narrative review, an omission is the lack of a discussion that deals with the vermiform appendix and the development of mucosal immunity, the appendix's biofilm and the role that an appendectomy may have in the development of RA (see PLoS One 2015;10, e0126816).
Author Response
Dear Reviewer,
we would like to thank you for your suggestion.
As you will see, we added a discussion about the role of vermiform appendix in the RA pathogenesis and in the gut dysbiosis (lines 448-461)
Moreover, we provided a spell checking of the whole manuscript
All the best
The Authors
Reviewer 2 Report
The authors summarized the relation of mucosal-environment with RA pathogenesis. This review paper is well-written of the present knowledge about RA pathogenesis. But this review has a few numbers of concerns need to be addressed by authors.
1. In “Lung as site of ACPAs production” section, not all person who have been exposed by nicotine or organic dusts developed arthritis. If there are compound risk factors with lung environment, the authors should write in this section.
2. Regarding line 267-272, if the profile of microbiota in healthy subjects was investigated in the study, the difference with healthy subjects should be discussed.
3. This review almost focused microbiota and ACPA production. The other evidence that microbiota affects to arthritis such as via cytokine production should be discussed, if possible.
4. There are some grammar mistakes. Page 4 line 121 “, although” should be corrected. Similally, page 2 line 91, “I” should be omitted.
Author Response
Dear Reviewer,
we would like to thank you for your suggestions.
As follows, the modifications of the paper that we performed:
1- We added a discussion about the compound risk of enviroment factors for RA mainly through the illustration of the gene-environment interaction, using the differences in the pathogenesis of seropositive versus seronegatives RA as example (lines 166-176)
2- We reported lso the microbiota profile of healthy subjects found in that study, relinking in the text to another paraghraper where we already discussed about the protective role of the healthy subjects' bacterial species (lines 281-284)
3- We integrated the discussion about the influence of microbiota on cytokines production already present in the manuscrip (lines 302-330, 433-445, 476-483), including a addictional part in which we summarize the cytokines produced at gut level and reporting a study about how gut microbiota influence systemic production of cytokines (lines 484-493)
4- We provided an estensive spell check of the whole manuscript
All the best
The Authors